# Physiological Doses of Oleic and Palmitic Acids Protect Human Endothelial Cells from Oxidative Stress

**DOI:** 10.3390/molecules27165217

**Published:** 2022-08-16

**Authors:** Olga M. Palomino, Veronica Giordani, Julie Chowen, Soledad Fernández Alfonso, Luis Goya

**Affiliations:** 1Faculty of Pharmacy, Universidad Complutense de Madrid, 28040 Madrid, Spain; 2L’Università di Urbino Carlo Bo, 61029 Urbino, Italy; 3Centro de Investigación Biomédica en Red: Fisiopatología de la Obesidad y Nutrición (CIBEROBN), Department of Endocrinology, Instituto de Investigación la Princesa, IMDEA Food Institute, CEI UAM + CSIC, Hospital Infantil Universitario Niño Jesús, 28009 Madrid, Spain; 4Department of Metabolism and Nutrition, Institute of Science and Food Technology and Nutrition (ICTAN—CSIC), 28040 Madrid, Spain

**Keywords:** EA.hy926, free fatty acids, reactive oxygen species, cardiovascular disease

## Abstract

Oxidative stress has been proposed to be a pathogenic mechanism to induce endothelial dysfunction and the onset of cardiovascular disease. Elevated levels of free fatty acids can cause oxidative stress by increasing mitochondrial uncoupling but, at physiological concentrations, they are essential for cell and tissue function and olive oil free fatty acids have proved to exhibit beneficial effects on risk factors for cardiovascular disease. We hypothesize that realistic concentrations within the physiological range of oleic (OA) and palmitic (PA) acids could be beneficial in the prevention of oxidative stress in vascular endothelium. Hence, pre-treatment and co-treatment with realistic physiological doses of palmitic and oleic acids were tested on cultured endothelial cells submitted to a chemically induced oxidative stress to investigate their potential chemo-protective effect. Cell viability and markers of oxidative status: reactive oxygen species (ROS), reduced glutathione (GSH), malondialdehyde (MDA), glutathione peroxidase (GPx) and glutathione reductase (GR) were evaluated. As a conclusion, the increased ROS generation induced by stress was significantly prevented by a pre- and co-treatment with PA or OA. Moreover, pre- and co-treatment of cells with FFAs recovered the stress-induced MDA concentration to control values and significantly recovered depleted GSH and normalized GPx and GR activities. Finally, pre- and co-treatment of cells with physiological concentrations of PA or OA in the low micromolar range conferred a substantial protection of cell viability against an oxidative insult.

## 1. Introduction

Endothelial dysfunction is a characteristic phenotype of cardiovascular complications involved in the pathogenesis of atherosclerosis, diabetes and related metabolic disorders [1]. Factors such as oxidative stress, hyperglycaemia, hyperlipidaemia and increased pro-inflammatory cytokines [1,2,3,4] contribute independently and/or synergistically to endothelial dysfunction by increasing endothelial inflammation and decreasing endothelial cell nitric oxide (NO) bioavailability. Endothelial cells constitute the innermost layer of blood vessels, so damage to endothelial cells is usually considered as the initial step of endothelial dysfunction. Further, endothelial dysfunction contributes to pathological processes specific to large (atherosclerosis) and small (retinopathy, nephropathy and neuropathy) blood vessels [1,5].

The vascular endothelium is continuously exposed to free fatty acids (FFAs), essential sources of energy within the cells released from circulating triglycerides by the lipoprotein lipase. They undergo beta-oxidation serving to ATP synthesis into mitochondria. Among the main FFAs, saturated non-esterified fatty acid palmitic acid (16:0, PA) and monounsaturated fatty acid oleic acid (18:1 n − 9, OA) are the most abundant [6]. Elevated FFAs have been shown to provoke vascular endothelial dysfunction whereas, when included in triglycerides from VLDL, they do not affect endothelium-dependent relaxation [7]; thus, increased circulating plasma FFA levels are associated with hypertriglyceridemia, diabetes or obesity [8,9]. In line with these findings, it has been proposed that expanded adipose tissue, due to its secretory output, is a strong risk factor for the development of cardiovascular diseases. In fact, the combined elevated release of fatty acids and adipokines by adipose tissue in obesity might be a link between adipose dysfunction, vascular inflammation and the development of atherosclerosis [10].

Oxidative stress has been proposed to be a potential pathogenic mechanism linking obesity and insulin resistance with endothelial dysfunction [11,12]. Overproduction of reactive oxygen species (ROS) disrupts the balance between oxidant and antioxidant defense systems. The oxidant systems are composed of NADPH oxidase, xanthine oxidase and lipid oxidation product malondialdehyde (MDA), meanwhile antioxidant systems include reduced glutathione (GSH), glutathione peroxidase (GPx), glutathione reductase (GR), catalase (CAT) and superoxide dismutase (SOD) [13]. Thus, attenuation of ROS is a strategy to deal with high-fat diet-induced vascular injury [4,14]. Although it has been widely reported that elevated levels of FFAs, in particular PA (the most abundant saturated fatty acid in the human body) and OA, can cause oxidative stress due to increased mitochondrial uncoupling [14,15,16], physiological concentrations of FFA are essential for cell and tissue function and, in particular, olive oil monounsaturated fatty acids (MUFA) have been proven to exhibit beneficial effects not only on primary cardiovascular disease (CVD) prevention, but also on several risk factors [17,18]. Therefore, we hypothesize that, contrary to the atherosclerotic effect of supra-physiological doses of essential fatty acids, realistic concentrations within the physiological range of OA and PA could be beneficial in the prevention of oxidative stress in vascular endothelium. To this purpose, a human endothelial cell line, EA.hy926, was used as a cell culture model of endothelium and treatment with a strong pro-oxidant, tert-butylhydroperoxide (t-BOOH), was used to reproduce an in vitro condition of oxidative stress in order to study the possible protective mechanisms through which essential free fatty acids protect endothelial function.

## 2. Results

### 2.1. ROS Production

As observed in the results of fluorescence assay depicted in Figure 1A, no significant increase in ROS generation was found in cultured EA.hy926 cells after treatment with 0.1 to 0.25 µM of PA or OA for 22 h, suggesting that cells so treated are not in a condition of oxidative stress. However, the highest doses tested of both compounds, 0.5 µM, evoked a significant increase of ROS levels indicating an apparent unbalance of the cell redox status after this long-term treatment. A pro-oxidant such as t-BOOH can directly oxidize DCFH to fluorescent DCF, and it can also decompose to peroxyl radicals and generate lipid peroxides and ROS, thus increasing fluorescence. When these endothelial cells in culture were treated with 100 µM t-BOOH for 22 h or 200 µM t-BOOH for 4 h, the two-fold increase of intracellular ROS concentration was indicative of a clear situation of oxidative stress (Figure 1B,C). However, this dramatic rise of ROS levels was significantly reduced when endothelial cells were pre-treated with tested concentrations of both fatty acids for 18 h prior to the oxidative challenge with 200 µM t-BOOH for 4 h (Figure 1B), and in both cases, chemo-protective response against ROS increase was dose-dependent. A similar dose-dependent reduction of t-BOOH-induced ROS over-production was observed when cultured endothelial cells were co-treated for 22 h with 100 µM t-BOOH and either PA or OA (Figure 1C).

### 2.2. GSH Concentration

Treatment of cultured EA.hy926 cells with 0.1 to 0.5 µM of OA for 22 h did not induce any significant change on basal concentration of GSH when compared to the Control value of 28.59 nmol/mg protein, as observed in Figure 2A. However, the same concentrations of PA evoked a dose-dependent increase on intracellular concentration of GSH (Figure 2A), suggesting a specific induction of PA on GSH through an increased synthesis or a decreased cellular degradation or depletion. When EA.hy926 cells were submitted to a condition of oxidative stress by the administration of 200 µM t-BOOH for 4 h or 100 µM t-BOOH for 22 h, a significant decrease of GSH between 20 to 80%, respectively, was observed (Figure 2B,C). This dramatic decrease of GSH was effectively prevented by pre-treating or co-treating endothelial cells with all tested doses of PA and OA except in the case of pre-treatment with 0.5 µM PA (Figure 2B,C). This result unequivocally indicates that the presence of these fatty acids in the culture media protects endothelial cells against the loss of reducing power in a situation of oxidative stress.

### 2.3. GPx Activity

No significant changes in the steady-state activity of GPx were observed when EA.hy926 cells in culture were treated with 0.1–0.5 µM PA for 24 h when compared with the Control value of 94.96 mU/mg protein (Figure 3A). However, treatment of EA.hy926 cells with similar concentrations of OA evoked a clear enhancement of GPx activity (Figure 3A), suggesting a specific inducing effect on this enzyme. When endothelial cells were challenged with 200 µM t-BOOH for 4 h or 100 µM t-BOOH for 22 h, a significant increase in the activity of this antioxidant defense was observed as a logic response to the induced ROS overproduction (Figure 3B,C). Pre-treatment of endothelial cells with all three doses of OA for 18 h before the onset of the condition of oxidative stress resulted in a dose-dependent reduction of GPx activity to reach basal values with 0.1 and 0.25 µM and below controls with 0.5 µM OA (Figure 3B). However, pre-treatment with any concentration of PA was inefficient to return GPx activity to values prior the stress (Figure 3B). On the other hand, co-treatment of cells with t-BOOH for 22 h plus 0.1 µM PA or 0.5 µM OA was capable of significantly reducing GPx activity whereas the rest of PA and OA doses were ineffective to recover the enhanced GPx activity (Figure 3C).

### 2.4. GR Activity

Similar to what was observed in GPx, treatment of EA.hy926 cells with all three concentrations of OA evoked a significant dose-dependent increase of GR activity when compared with the control value of 3.72 mU/mg protein (Figure 4A), suggesting a specific stimulating effect on this enzyme. Moreover, as in the case of GPx, neither dose of PA evoked a significant change of GR activity in a direct treatment (Figure 4A), but when EA.hy926 cells in culture were submitted to a challenge with 200 µM t-BOOH for 4 h or 100 µM t-BOOH for 22 h a remarkable 70% increase in the GR activity was detected as a physiological response to the increase of GSSG produced by the high activity of GPx (Figure 4B,C). A significant recovery of the GR activity to unstressed values was observed when endothelial cells were pretreated with 0.5 µM PA or with the three OA concentrations (Figure 4B); similarly, all tested doses except 0.1 µM OA were able to significantly decrease GR activity in the co-treatment assay (Figure 4C).

### 2.5. MDA Concentration as Index of Lipid Peroxidation

Since no significant effect on steady-state MDA was expected after direct treatment of endothelial cells with the physiological concentrations of PA and OA, only protective tests of pre- and co-treatment assays were performed for this parameter. Table 1 shows that an increase of over 60% in MDA concentration was found in endothelial cells after treatment with 100 µM t-BOOH for 22 h or 200 µM t-BOOH for 4 h. All doses of PA and OA significantly restrained the t-BOOH-induced raise of MDA both in co-treatment and pre-treatment assays. Remarkably, in most cases values of MDA remained below those of control untreated cells, especially in the co-treatment condition, suggesting an evident restriction of lipid peroxidation by these physiological doses of the two necessary fatty acids.

### 2.6. Cell Viability

Treatment of cultured EA.hy926 with PA or OA concentrations ranging from 0.1 to 0.5 µM for 22 h did not evoke any significant cell damage; even a slight increase in cell number was observed with OA, as shown in the results of the crystal violet assay depicted in Figure 5A. On the contrary, treatment of cells with 100 µM t-BOOH for 22 h or 200 µM for 4 h provoked a substantial decrease of cell viability of 20 and 40%, respectively (Figure 5B,C). This dramatic cell damage was completely avoided when cells were pre-treated with any PA and OA concentration for 18 h before the challenge with the pro-oxidant (Figure 5B), indicating an absolute protection of endothelial cell viability. Similarly, a significant reduction of cell death induced by t-BOOH was observed when cultured cells were submitted to simultaneous co-treatment with PA or OA and pro-oxidant for 22 h (Figure 5C), although in this case, only a partial chemo-protection was observed. Cell viability recovery against oxidative stress seems to be dose-dependent for OA (Figure 5C).

## 3. Discussion

Endothelial dysfunction is an early manifestation common to several cardiovascular risk factors such as arterial hypertension, arteriosclerosis, smoking, diabetes, or aging [2,19,20]. It is now widely assumed that elevated concentrations of FFAs are involved in the development of a chronic low-grade inflammatory state that contributes to vascular dysfunction thereby providing a direct link between obesity and vascular complications [21,22]. However, physiological concentrations of FFA are essential for cell and tissue nutrition and function as well as on primary CVD prevention [17,18]; the hypothesis of this study can be accepted, as a significant chemo-protective effect of physiological doses of PA and OA against an induced oxidative stress is shown for the first time in endothelial cells. 

Dietary fats are important factors that determine serum lipid concentrations [23]; indeed, administration to animals or treatment of cultured cells with elevated FFAs doses, especially PA, has been widely utilized as experimental models of endothelial dysfunction through oxidative stress. Thus, doses of 300 µM PA and higher have been reported to induce oxidative stress and endothelial function impairment [14,24,25,26]. Similarly, concentrations over 300 µM OA resulted in a significant increase in ROS production in endothelial cells [3,27]. Furthermore, smaller doses of 60 µM OA were used to induce oxidative stress in ECV-304 cells in combination with high glucose [28] and a combination of both OA and PA at a final concentration of 100 µM were applied to human vascular smooth muscle cells to evoke endothelial damage [22]. Finally, 180.6 nM and 20.3 nM of unbound oleic and palmitic acids induced direct apoptosis in cultured endothelial cells [29]. Regular values of FFAs in human blood range from 5–10 nM with slight increase after overnight fasting [30]. Interestingly, these values now widely accepted were as much as 1000-fold smaller than values previously estimated. Thus, our treatment with concentrations represents about 100-fold the standard values, but still 100-fold lower than doses that have been reported to induce oxidative stress and endothelial function impairment.

Since oxidative stress has proved to be an intrinsic pathogenic mechanism linking high serum FFAs and CVD [11,31,32], prevention or attenuation of pro-oxidant conditions in endothelial cells could be a valid strategy to prevent or restrain vascular dysfunction [1,4,14,33]. EA.hy926 is a permanent endothelial cell line derived from human umbilical vein endothelial cells that has proved a reliable model of endothelial tissue to test the effect of natural products [34,35,36]. In the present study, EA.hy926 viability was not altered by treatment with concentrations up to 0.5 µM palmitic or oleic acids for 22 h, indicating no endothelial cell toxicity in basal conditions. It is crucial to ensure that no direct cell damage is caused by the antioxidant before testing the potential chemo-protective capacity since, as stated above, elevated doses of FFAs may also act as pro-oxidants in cell culture systems and evoke cellular damage [14]. Our results agree with previous studies reporting no toxic effects of PA and OA on EA.hy926 cells [9,37]. Once ascertaining that doses of PA and OA selected were safe, the same doses were tested to protect cells from an oxidative insult. 

A reliable assessment of an oxidative stress condition in living cells can be achieved by direct evaluation of the ROS produced by the cell mitochondria [38]. Reduced ROS generation has been reported in cells treated with natural polyphenols and other phytochemicals with antioxidant capacity due to their chemical structure [39]; however, molecules of fatty acids such as PA and OA do not have substantial antioxidant or radical quenching potential, therefore, no direct effect on ROS production was expected. Consequently, plain treatment of EA.hy926 cells with PA or OA did not affect basal ROS generation, except for the highest dose of both fatty acids, which seem to be stringent enough to evoke a significant response of ROS. Nevertheless, co-treatment or pre-treatment with all concentrations of PA and OA were able to significantly decrease the dramatic ROS overproduction induced by the pro-oxidant t-BOOH to levels that were either similar to those of controls or at least more manageable by the cells. These results clearly indicate that intensified values of ROS generated during the stress period are being more competently extinguished in cells co-treated or pre-treated with PA or OA certainly resulting in a reduced cell oxidative damage.

As the main endogenous antioxidant defense within the cell, it is established that GSH depletion reflects an environment of intracellular oxidation, whereas an increase in GSH concentration places the cell in an advantageous position against a potential oxidative insult [38]. In our experimental conditions, most of PA and OA doses recovered the dramatic decrease in the cell GSH pool induced by a stressful situation. A similar recovery of depleted GSH has been previously reported in the same cell line treated with polyphenolic extracts from Silybum marianum fruit [37], Vochysia rufa stem bark [35] and cocoa flavanols [36], as well as in other cell lines mostly with pure antioxidants and polyphenolic extracts [38]. This protective effect on the cellular antioxidant stores is essential since maintaining GSH concentration above a critical threshold while facing a stressful situation represents, together with the reduced ROS production reported above, an enormous benefit for cell survival.

Activation of GPx and GR is an essential mechanism of the cell antioxidant defense system to face oxidative challenges and consequently plays a crucial role to reduce ROS production in the presence of t-BOOH [34,35,36]. An enhanced GPx activity fights ROS overproduction at the expense of GSH which oxidizes to GSSG that is recovered back to GSH by GR; however, a rapid return of the antioxidant enzyme activities to basal values once the oxidative condition has been surpassed will reinstate favorable conditions for the cell to cope with a new oxidative insult. In EA.hy926 cells an increase of the activity of GPx and GR in response to the oxidative challenge induced by t-BOOH was significantly restricted when cells were co-treated or pre-treated with the essential fatty acids. The decrease of the activity of GPx and GR when PA or OA were added to the cells in stressful conditions might suggest a decreased necessity for the endogenous antioxidant defense system in the presence of these essential FFAs; henceforth, while cells submitted to oxidative stress are still struggling to overcome the insult, those treated with the fatty acids have capably controlled the stressful environment and returned to a balanced redox status.

MDA, a three-carbon compound formed by scission of peroxidized polyunsaturated fatty acids, is widely used as an index of lipo-peroxidation in biomedical sciences [40]. MDA has been found raised in several diseases related to free radical injury; thus, the significant protection by two essential fatty acids, OA and PA, against an induced lipid peroxidation in EA.hy926 cells indicates a genuine approach in the prevention of oxidative stress-derived pathologies such as endothelial dysfunction and CVD. A similar chemo-protection against lipid peroxidation has been recently reported in the same cell culture model with phenolic extracts from *Silybum marianum* fruit [34], *Vochysia rufa* stem bark [35] and cocoa powder [36] that contain high amounts of natural antioxidants.

Since both PA and OA seem to regulate the antioxidant defense mechanisms necessary to manage the oxidative challenge, their protective effect should be evidenced in higher cell viability. A potent oxidative challenge induced t-BOOH severely compromised cell viability but a simultaneous co-treatment with any dose of fatty acids completely avoided the alleged cell damage as well as pre-treatment of cells with PA or OA also showed a significant recovery of cell viability indicating that integrity of cells was substantially protected against the oxidative insult. Other authors have found no protective effect of 25–50 µM OA and PA against TNF-alpha induced damage [37], but a significant protective effect has been previously reported in cultured endothelial cells treated with plant antioxidants [34,35,36,41,42,43,44], and this is the first report, to our knowledge, of an endothelial cell viability protection against an oxidative damage by physiological doses of two necessary fatty acids. It is worth noting that a chemo-protective effect in a pre-treatment assay is expected to be more efficient than in a co-treatment model, since in the latter the continuous presence of the pro-oxidant in the culture media (although in a lower concentration) results in a more stringent situation. In accordance, a clearer dose-dependent response for ROS, GPx, GR, MDA and cell viability was generally observed in the pre-treatment than in the co-treatment condition. However, it should not be forgotten that a co-treatment condition, where both pro-oxidants and antioxidants are concurrently in our tissues, is closer to the physiological in vivo situation. Since these compounds do not show any antioxidant capacity, the intrinsic mechanism of PA and OA to provide protection to the endothelial cells in order to endure an oxidative challenge is still unclear, even more so considering that concentrations as low as 180.6 nM OA and 20.3 nM PA provoked direct apoptosis in cultured endothelial HUVEC cells [29]. Nevertheless, incorporation of them in their membranes can have an important impact on cells biology [45]; thus, improvement of endothelial dysfunction by polyunsaturated fatty acids (PUFA) has been reported to be mediated by reducing expression of adhesion molecules and restoring NO bioavailability [46,47]. However, scarce information is available regarding PA and OA; thus, oleates and olive oil polar phenolic compounds, either individually, as shown by E-selectin reduction, or synergistically, as shown by vascular cell adhesion molecule 1 (VCAM-1) decrease, may improve endothelial function in the aorta of the cholesterol-fed rats [17]. Also, very recently, 20 µM PA and OA reduced expression of Intercellular Adhesion Molecule 1 (ICAM-1) in endothelial cells stimulated with tumor necrosis factor (TNF)-alpha [35]. The study of inflammatory factors, nitric oxide concentration and endothelial NOS will surely provide relevant information to unravel the protective role of PA and OA [48]. 

A tempting comparison of chemo-protective capacity of both fatty acids in cultured endothelial cells could indicate that PA and OA had a comparable effect in ROS and MDA levels; PA was better in co-treatment and OA better in pre-treatment for GSH concentration; whereas OA showed better results in GPx and GR activity as well as cell viability.

As a main conclusion of this study, the protective mechanism of physiological concentrations of PA and OA on EA.hy926 cells submitted to an oxidative stress could be explained in terms of regulation of the cellular redox status; accordingly, pre- or co-treatment of cultured cells with OA or PA decreases ROS production during stress and reduces the need of peroxide detoxification through GPx as well as of GSH, resulting in diminished cell injury and death. Concentrations of OA and PA in the nanomolar-low micromolar range tested in the present study are far from the systemic levels necessary to evoke endothelial damage and increase the atherogenic index; but OA/PA doses added to the cultures are still well above the nanomolar levels usually found in post-prandial blood, which would make interesting to investigate the potential implication of the FFA treatment on the lipid metabolic profile regarding mitochondrial OxPhos and cAMP/PKA signaling. Nevertheless, in this first study on the effect of main FFA on endothelial function we have only addressed the cell antioxidant defense response to oxidative stress. So far, the present data propose a prominent role for physiological doses of necessary FFAs in the protection afforded against pathologies such as CVD and diabetes, for which excessive production of ROS has been implicated as a causal or contributory factor.

## 4. Materials and Methods

### 4.1. Reagents

Tert-butylhydroperoxide (t-BOOH), GR, reduced and oxidized (GSSG) glutathione, di-chlorofluorescin (DCFH), o-phthaldialdehyde (OPT), nicotine adenine dinucleotide phosphate reduced salt (NADPH), 2,4-dinitrophenylhydrazine (DNPH), H2O2, 1,1,3,3-tetraethoxypropane (TEP), gentamicin, penicillin G and streptomycin were purchased from Sigma Chemical Co. (Madrid, Spain). Acetonitrile, methanol of HPLC grade, dimethyl sulfoxide (DMSO) of analytical grade and all other usual laboratory reagents were acquired from Panreac (Barcelona, Spain). Bradford reagent was from BioRad Laboratories S.A. DMEM culture media and fetal bovine serum (FBS) were from Cultek (Madrid, Spain). All other reagents were of analytical quality.

### 4.2. Sample Preparation

Fatty acid stock solutions of 200 mM were prepared in 100% EtOH. Working solutions of 1 mM fatty acids were made by incubating the fatty acids in media containing 10% endotoxin and fatty acid free BSA at 37 °C for 30–60 min with occasional vortexing. This solution was then added to cells to obtain the final fatty acid concentrations. The fatty acid-albumin molar ratio was kept at <3 to ensure that the fatty acids were bound to albumin and equal volumes of the medium/EtOH/BSA vehicle were applied to control cells [49].

### 4.3. Cell Culture

EA.hy926, a human hybrid cell line, was a kind gift from Profs. Patricio Aller and Carmelo Bernabeu, Centro de Investigaciones Biológicas, CSIC, Madrid, Spain. The cell line was cultured and passaged in Bio-Whittaker DMEM media supplemented with 10% fetal bovine serum. Cells were maintained in a humidified incubator containing 5% CO_2_ and 95% air at 37 °C and grown in DMEM medium supplemented with 10% FBS and 50 mg/L of each of the following antibiotics: gentamicin, penicillin and streptomycin. The culture medium was changed every other day in order to remove the non-adherent and dead cells, and the plates were usually split 1:3 when they reached confluence.

### 4.4. Cell Treatment

Different concentrations of PA (0.1, 0.25 and 0.5 µM) and OA (0.1, 0.25 and 0.5 µM), dissolved from the 1 mM stock solution in serum-free culture medium, were added to the cell plates for 22 h to study a direct/basal effect of the compounds. In order to evaluate the protective effect of PA and OA against an oxidative insult, two different approaches were carried out, co-treatment and pre-treatment. In the co-treatment assay EA.hy926 cells were simultaneously treated for 22 h with 100 µM t-BOOH plus any of the four different PA or OA concentrations, whereas in the pre-treatment assay cells were first treated with tested doses of PA or OA for 18 h, then washed and submitted to a new media containing 200 µM t-BOOH for 4 h.

### 4.5. Determination of ROS

Cellular ROS were quantified by the DCFH assay using microplate reader with slight modifications [34,50]. For the assay, cells were seeded in 24-well plates at a rate of 2 × 105 cells per well and changed to the different PA and OA concentrations the day after. Prior to the end of the assay, 5 µM DCFH was added to the wells for 30 min at 37 °C. Then, cells were washed twice with serum-free medium before multiwell plates were measured in a fluorescent microplate reader at excitation wavelength of 485 nm and emission wavelength of 530 nm. After being oxidized by intracellular oxidants, DCFH will become dichlorofluorescein (DCF) and emit fluorescence. By quantifying fluorescence over a period of 90–120 min, a fair estimation of the overall oxygen species generated under the different conditions was obtained. This parameter gives a very good evaluation of the degree of cellular oxidative stress. The assay has been described elsewhere [34,50].

### 4.6. Determination of GSH Concentration

The content of GSH was quantitated by the fluorometric assay described in Browne & Armstrong [51] with some modifications. The method takes advantage of the reaction of GSH with OPT at pH 8.0; although it has been observed that OPT reacts not only with GSH but also with other thiols, such as cysteine and N-acetylcysteine, comparison to appropriate controls permitted a reliable quantification. Values of GSSG were always below 10% of total cellular glutathione and slight variations were observed. After the different treatments, the culture medium was removed, and cells (4 × 106) were detached and homogenized by ultrasound with 5% trichloroacetic acid containing 2 mM EDTA. Following centrifugation of cells for 30 min at 1000× *g*, 50 µL of the clear supernatant were transferred to a 96 multiwell plate for the assay. Fluorescence was measured at an excitation wavelength of 345 nm emission wavelength of 425 nm. The results of the samples were referred to those of a standard curve of GSH. 

### 4.7. Determination of GPx and GR Activity

For the assay of the GPx and GR activity, treated cells (4 × 106) were suspended in PBS and centrifuged at 300 g for 5 min to pellet cells. Cell pellets were resuspended in 20 mM Tris, 5 mM EDTA and 0.5 mM mercaptoethanol, sonicated and centrifuged at 3000× *g* for 15 min. Enzyme activities were measured in the supernatants. Determination of GPx activity is based on the oxidation of GSH by GPx, using t-BOOH as a substrate, coupled to the disappearance of NADPH by GR as described in Alía et al. [50] with slight modifications. GR activity was determined by following the decrease in absorbance due to the oxidation of NADPH utilized in the reduction of oxidized glutathione [34,50]. Protein was measured by using the Bradford reagent.

### 4.8. Determination of MDA Concentration

Cellular MDA was analyzed by high-performance liquid chromatography (HPLC) as its DNPH derivative [52]. Briefly, treated cells (6 × 106) were collected in PBS and centrifuged at 220× *g* for 5 min at 4 °C. Then, the pellet was suspended in 200 µL PBS and sonicated. After centrifugation at 3500× *g* for 15 min, 125 µL of cytoplasmatic content was mixed with 25 µL of 6 M sodium hydroxide and incubated in a 60 °C water bath for 30 min, to achieve alkaline hydrolysis of protein bound MDA. Protein was precipitated by adding 62.5 µL of 35 % (*v*/*v*) perchloric acid and the mixture was centrifuged at 2800× *g* for 10 min. A volume of 125 µL of supernatant was mixed with 12.5 µL DNPH prepared as a 5 mM solution in 2M hydrochloric acid. Finally, this reaction mixture was injected onto an Agilent 1100 Series HPLC-DAD. Quantification of MDA as the major end product of lipid peroxidation was carried on by chromatographic evaluation by HPLC according to Mateos et al. [52]. In brief, a sample aliquot of 20 µL was injected onto an Agilent 1260 HPLC–DAD with a Fortis C18 column (4.6 mm × 250 mm, 5 µm particle size, Fortis Technologies Ltd., Neston, UK) and isocratically eluted with 1% (*v*/*v*) acetic acid in deionized water and acetonitrile (62:38, *v*/*v*) at a flow rate of 0.6 mL/min. The MDA-TBA adduct was detected at 310 nm. Quantification was calculated from the area, based on a calibration chromatogram performed with a standard solution of MDA prepared by acid hydrolysis as previously described [51]. MDA values are expressed as nmol of MDA/mg protein; protein was measured by using the Bradford reagent.

### 4.9. Evaluation of Cell Viability

Cell viability was determined by using the crystal violet assay [53]. Cells were seeded at low density (1 × 104 cells per well) in 96-well plates, grown for 18 h and incubated with crystal violet (0.2% in ethanol) for 20 min. Plates were rinsed with water and 1% sodium dodecylsulphate added. The absorbance of each well was measured using a microplate reader at 570 nm.

### 4.10. Statistical Analyses

Statistical analysis of data was as follows: prior to analysis, the data were tested for homogeneity of variances by the test of Levene; for multiple comparisons, one-way ANOVA was followed by a Bonferroni test when variances were homogeneous or by Tamhane test when variances were not homogeneous. The level of significance was *p* < 0.05. A SPSS version 23.0 program has been used. Different small letters upon data bars or as super index indicate statistically significant differences (*p* < 0.001) among the different groups or conditions; data sharing a statistical letter are not significantly different.

## Figures and Tables

**Figure 1 molecules-27-05217-f001:**
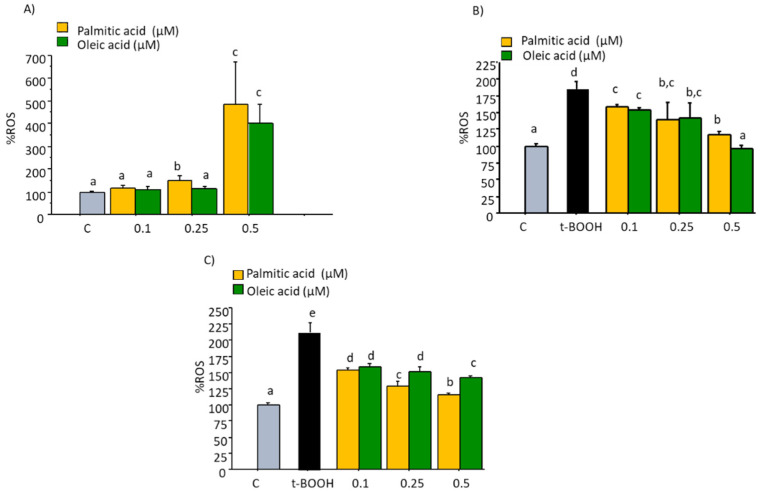
Effect of Palmitic acid (PA) and Oleic acid (OA) on ROS production by EA.hy926 cells. (**A**) direct effect of PA and OA; (**B**) effect of pre-treatment of noted PA or OA concentrations for 18 followed by 4 h with 200 µM t-BOOH; (**C**) effect of co-treatment of 100 µM t-BOOH plus noted PA or OA concentrations for 22 h. Results are means ± SD (*n* = 4 replicates). Within each panel, different letters upon data bars indicate significant differences (*p* < 0.05) among data.

**Figure 2 molecules-27-05217-f002:**
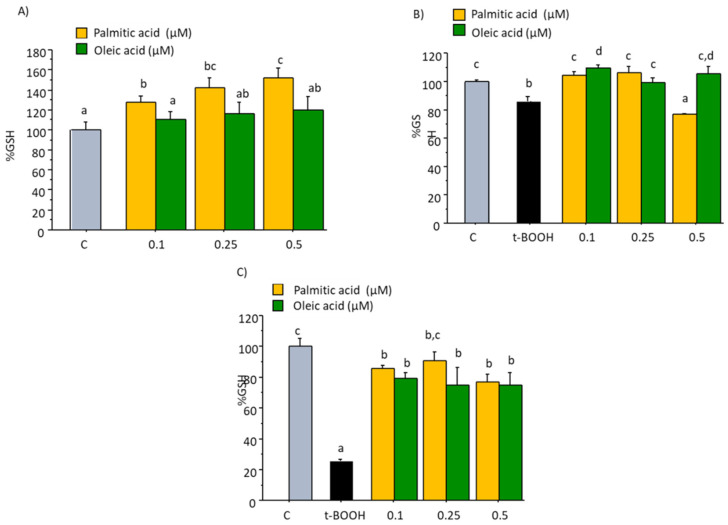
Effect of palmitic acid (PA) and oleic acid (OA) on GSH concentration in EA.hy926 cells. (**A**) direct effect of PA and OA; (**B**) effect of pre-treatment of noted PA or OA concentrations for 18 followed by 4 h with 200 µM t-BOOH; (**C**) effect of co-treatment of 100 µM t-BOOH plus noted PA or OA concentrations for 22 h. Results are means ± SD (*n* = 4 replicates). Within each panel, different letters upon data bars indicate significant differences (*p* < 0.05) among data.

**Figure 3 molecules-27-05217-f003:**
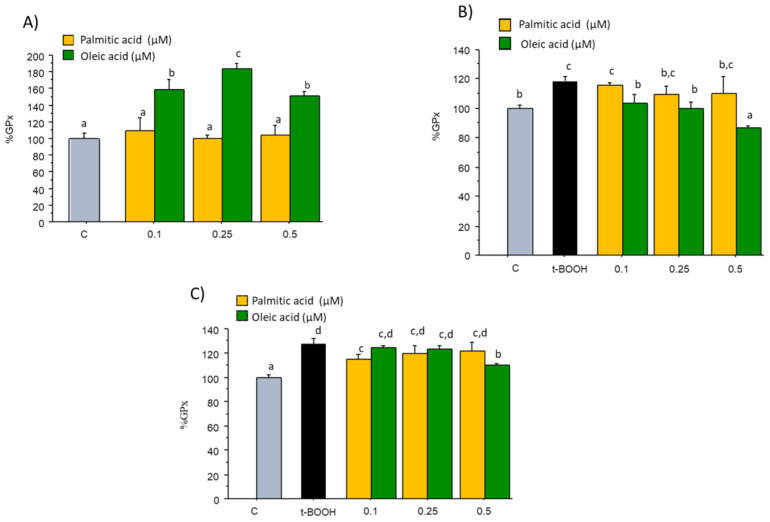
Effect of palmitic acid (PA) and oleic acid (OA) on GPx activity in EA.hy926 cells. (**A**) direct effect of PA and OA; (**B**) effect of pre-treatment of noted PA or OA concentrations for 18 followed by 4 h with 200 µM t-BOOH; (**C**) effect of co-treatment of 100 µM t-BOOH plus noted CPE or EC concentrations for 22 h. Results are means ± SD (*n* = 3–4 replicates). Within each panel, different letters upon data bars indicate significant differences (*p* < 0.05) among data.

**Figure 4 molecules-27-05217-f004:**
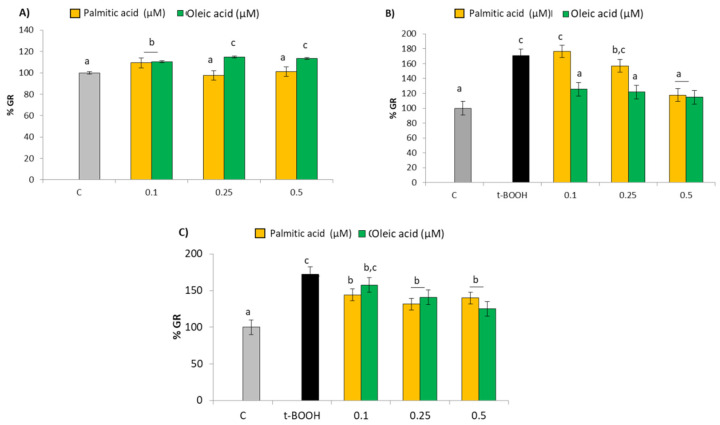
Effect of palmitic acid (PA) and oleic acid (OA) on GR activity in EA.hy926 cells. (**A**) direct effect of PA and OA; (**B**) effect of pre-treatment of noted PA or OA concentrations for 18 followed by 4 h with 200 µM t-BOOH; (**C**) effect of co-treatment of 100 µM t-BOOH plus noted PA or OA concentrations for 22 h. Results are means ± SD (*n* = 3–4 replicates). Within each panel, different letters upon data bars indicate significant differences (*p* < 0.05) among data.

**Figure 5 molecules-27-05217-f005:**
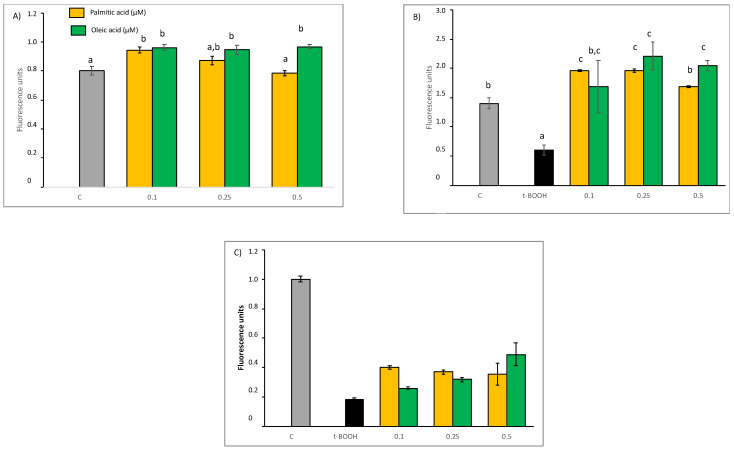
Effect of palmitic acid (PA) and oleic acid (OA) on EA.hy926 cell viability. (**A**) direct effect of PA and OA; (**B**) effect of pre-treatment of noted PA or OA concentrations for 18 followed by 4 h with 200 µM t-BOOH; (**C**) effect of co-treatment of 100 µM t-BOOH plus noted PA or OA concentrations for 22 h. Results are means ± SD (*n* = 8 replicates). Within each panel, different letters upon data bars indicate significant differences (*p* < 0.05) among data.

**Table 1 molecules-27-05217-t001:** Effect of co-treatment and pre-treatment of EA.hy926 cells with noted concentrations of palmitic acid (PA) and oleic acid (OA) on MDA concentration expressed as nmol/mg protein. Results are means ± SD (*n* = 3–4 replicates). Different letters in each column indicate statistically significant differences (*p* < 0.05) among data.

	nmol MDA/mg prot ± SD	
	**Cotreat**	**Pretreat**
**Control**	1.17 ^c^ ± 0.23	1.50 ^c^ ± 0.24
**t-BOOH**	1.96 ^d^ ± 0.31	2.46 ^d^ ± 0.39
**0.1 µM oleic acid**	0.50 ^b^ ± 0.04	0.25 ^a^ ± 0.03
**0.25 µM oleic acid**	0.46 ^b^ ± 0.15	1.15 ^b^ ± 0.31
**0.5 µM oleic acid**	0.53 ^b^ ± 0.07	1.15 ^b^ ± 0.04
**0.1 µM palmitic acid**	0.58 ^b^ ± 0.23	1.02 ^b^ ± 0.07
**0.25 µM palmitic acid**	0.62 ^b^ ± 0.08	0.86 ^b^ ± 0.09
**0.5 µM palmitic acid**	0.24 ^a^ ± 0.03	1.14 ^b^ ± 0.21

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
