# Peer review of "Physiological Doses of Oleic and Palmitic Acids Protect Human Endothelial Cells from Oxidative Stress"

_molecules, 2022, doi:10.3390/molecules27165217_

Round 1
Reviewer 1 Report
The paper describes beneficial effects of PA and OA on human endothelial cells subjected to oxidative stress. I have the following comments:
1. In all Figures the data are presented in % of a control value. While this makes sense for ROS production shown in Fig. 1, the remaining data would be more convincing, if absolute numbers would be shown. That problem is especially visible in Fig. 5, which contains cell viabilities above 100%. Here, definitely the absolute viability should be shown. For GSH, GPx and GR at least the value which is equivalent to 100% should be given.
2. Statistical assessment: Here in the methods part a notion is missing what the different letters on the bar graphs mean (what is the level of significance?).
3. It is not true to state that PA or OA are essential fatty acids (as made in line 194, line 375), since they can be synthesized by the fatty acid synthesis pathway. Essential fatty acids contain at least 2 non-conjugated double bonds.
4. The authors need to discuss in much more detail potential effects of the added FA's relevant for their findings, including potential metabolic effects, uncoupling of OxPhos etc..
Author Response
Reply to reviewer 1
We wish to thank the reviewer for the careful reading and constructive comments on the manuscript.
The paper describes beneficial effects of PA and OA on human endothelial cells subjected to oxidative stress. I have the following comments:
- In all Figures the data are presented in % of a control value. While this makes sense for ROS production shown in Fig. 1, the remaining data would be more convincing, if absolute numbers would be shown. That problem is especially visible in Fig. 5, which contains cell viabilities above 100%. Here, definitely the absolute viability should be shown. For GSH, GPx and GR at least the value which is equivalent to 100% should be given.
Reply: in agreement with the reviewer´s comment, we have made all changes recommended: % value of ROS has been maintained since, as stated by the reviewer, they show arbitrary data; cell viabilities in actual figures have been included in figure 5, and actual values for GSH, GPx and GR equivalent to 100 % (given to controls) have been included in the text within the results section.
- Statistical assessment: Here in the methods part a notion is missing what the different letters on the bar graphs mean (what is the level of significance?).
Reply: in agreement with the reviewer´s comment, the meaning of the different letters showing statistical significance has also been included in the Material and Methods section. Indeed, addition of different letters indicating statistical significance among data is highly recommended by most statisticians, including those responsible of the Statistic Centre at CSIC (Spanish Council of Scientific Research). Several years ago, we were encouraged by Dr. Laura Barrios at the mentioned center to present all our data adding different letters upon data bars to indicate significant differences among data when data showed different letters. In fact, I myself recommend this type of statistical representation in all my peer-reviews to other authors. However, we agree with the reviewer on the fact that this has to be properly explained and, after all these years, we have largely disregarded this description. The correct significance of the statistical letters, instead of symbols, has now been included in figure legends, as well as Material and Methods.
- It is not true to state that PA or OA are essential fatty acids (as made in line 194, line 375), since they can be synthesized by the fatty acid synthesis pathway. Essential fatty acids contain at least 2 non-conjugated double bonds.
Reply: In full agreement with the reviewer, the word essential has been changed for necessary. We wanted to emphasize the crucial role of OA and PA but it is true that the word essential should be restricted to metabolites/molecules that are not synthetized in our cells.
- The authors need to discuss in much more detail potential effects of the added FA's relevant for their findings, including potential metabolic effects, uncoupling of OxPhos etc.
Reply: we agree on the fact that the metabolic implications of the exogenous addition of OA and PA to our cell cultures are not widely discussed, but we felt that the goal of the study was more oriented to the redox status than to the metabolic situation and we did not dare to speculate too much on that area. Although it is true, as stated in discussion, that the tested concentrations (nanoM-microM range) were well above the usual blood values, it is also significant that concentrations above 200 microM (40-fold of the ones tested in the study) of any of the two FFA are needed to actually alter the metabolic profile and create a possible pathological condition. However, as suggested by the reviewer, we have pointed out the potential implication of the FFA treatment regarding OxPhos and a short sentence on this particular point has been now added to the text:
Concentrations of OA and PA in the nanomolar-low micromolar range tested in the present study are far from the systemic levels necessary to evoke endothelial damage and increase the atherogenic index; but OA/PA doses added to the cultures are still well above the nanomolar levels usually found in post-prandial blood, which would make interesting to investigate the potential implication of the FFA treatment on the lipid metabolic profile regarding mitochondrial OxPhos and cAMP/PKA signaling. Nevertheless, in this first study on the effect of main FFA on endothelial function we have only addressed the cell antioxidant defense response to oxidative stress.

Reviewer 2 Report
The paper molecules-1809285 describes the protective role of two common fatty acids, saturated palmitic acid and monounsaturated oleic acid in terms of cardiovascular complications (endothelial dysfunction) caused by oxidative stress.
The paper shows that both fatty acids at low concentrations possess the anti-oxidative capability, however with small differences in action.
The introduction is presented clearly, and so is the material and method section. However, the results need to be corrected according to the following:
1. What are the letters (a, b, c, d, etc) on the figures showing statistical significance? Why letters have been used and what is their meaning?
2. Describing results it would be good to see the p-value of a particular analysis, not just the increase or decrease. That would tell a lot about the significance.
3. Table 1. Instead of letters describing statistical significance what is not informative for the reader should be an additional column with p-value of each comparison.
In the discussion, it would be good to discuss the potential protective mechanism of both palmitic and oleic acids and the differences of action that have been presented in the results of this paper.
Author Response
We wish to thank the reviewer for the careful reading and constructive comments on the manuscript.
The paper molecules-1809285 describes the protective role of two common fatty acids, saturated palmitic acid and monounsaturated oleic acid in terms of cardiovascular complications (endothelial dysfunction) caused by oxidative stress.
The paper shows that both fatty acids at low concentrations possess the anti-oxidative capability, however with small differences in action.
The introduction is presented clearly, and so is the material and method section. However, the results need to be corrected according to the following:
- What are the letters (a, b, c, d, etc) on the figures showing statistical significance? Why letters have been used and what is their meaning?
- Describing results it would be good to see the p-value of a particular analysis, not just the increase or decrease. That would tell a lot about the significance.
- Table 1. Instead of letters describing statistical significance what is not informative for the reader should be an additional column with p-value of each comparison.
Reply to Q1-3: in agreement with the reviewer´s comment, the meaning of the different letters showing statistical significance has also been included in the Material and Methods section. Indeed, addition of different letters indicating statistical significance among data is highly recommended by most statisticians, including those responsible of the Statistic Centre at CSIC (Spanish Council of Scientific Research). Several years ago, we were encouraged by Dr. Laura Barrios at the mentioned center to present all our data adding different letters upon data bars to indicate significant differences among data when data showed different letters. In fact, I myself recommend this type of statistical representation in all my peer-reviews to other authors. However, we agree with the reviewer on the fact that this has to be properly explained and, after all these years, we have largely disregarded this description. The correct significance of the statistical letters, instead of symbols, has now been included in figure legends, as well as Material and Methods.
With respect to Table 3, a different letter in each column indicated a p value < 0.05, which means statistical difference among data.
In the discussion, it would be good to discuss the potential protective mechanism of both palmitic and oleic acids and the differences of action that have been presented in the results of this paper
Reply: we agree on the fact that the metabolic implications of the exogenous addition of OA and PA to our cell cultures are not widely discussed, but we felt that the goal of the study was more oriented to the redox status than to the metabolic situation and we did not dare to speculate too much on that area. Although it is true, as stated in discussion, that the tested concentrations (nanoM-microM range) were well above the usual blood values, it is also significant that concentrations above 200 microM (40-fold of the ones tested in the study) of any of the two FFA are needed to actually alter the metabolic profile and create a possible pathological condition. However, as suggested by the reviewer, we have pointed out the potential implication of the FFA treatment regarding OxPhos and a short sentence on this particular point has been now added to the text:
Concentrations of OA and PA in the nanomolar-low micromolar range tested in the present study are far from the systemic levels necessary to evoke endothelial damage and increase the atherogenic index; but OA/PA doses added to the cultures are still well above the nanomolar levels usually found in post-prandial blood, which would make interesting to investigate the potential implication of the FFA treatment on the lipid metabolic profile regarding mitochondrial OxPhos and cAMP/PKA signaling. Nevertheless, in this first study on the effect of main FFA on endothelial function we have only addressed the cell antioxidant defense response to oxidative stress.

Round 2
Reviewer 1 Report
A of my comments have been addressed appropriately.